# High Genetic Diversity and Virulence Potential in *Bacillus cereus sensu lato* Isolated from Milk and Cheeses in Apulia Region, Southern Italy

**DOI:** 10.3390/foods12071548

**Published:** 2023-04-06

**Authors:** Angelica Bianco, Giovanni Normanno, Loredana Capozzi, Laura Del Sambro, Laura Di Fato, Angela Miccolupo, Pietro Di Taranto, Marta Caruso, Fiorenza Petruzzi, Ashraf Ali, Antonio Parisi

**Affiliations:** 1Experimental Zooprophylactic Institute of Apulia and Basilicata, Via Manfredonia 20, 71121 Foggia, Italy; 2Department of Sciences of Agriculture, Food, Natural Resources and Engineering (DAFNE), University of Foggia, Via Napoli 25, 71122 Foggia, Italy

**Keywords:** *B. cereus*, MLST, food safety, dairy products

## Abstract

The *Bacillus cereus* group includes species that act as food-borne pathogens causing diarrheal and emetic symptoms. They are widely distributed and can be found in various foods. In this study, out of 550 samples of milk and cheeses, 139 (25.3%) were found to be contaminated by *B. cereus sensu lato* (*s.l.*). One isolate per positive sample was characterized by Multilocus Sequence Typing (MLST) and for the presence of ten virulence genes. Based on MLST, all isolates were classified into 73 different sequence types (STs), of which 12 isolates were assigned to new STs. Virulence genes detection revealed that 90% and 61% of the isolates harboured the *nheABC* and the *hblCDA* gene cluster, respectively. Ninety-four percent of the isolates harboured the enterotoxin genes *entS* and *entFM*; 8% of the isolates possessed the *ces* gene. Thirty-eight different genetic profiles were identified, suggesting a high genetic diversity. Our study clearly shows the widespread diffusion of potentially toxigenic isolates of *B. cereus s.l.* in milk and cheeses in the Apulia region highlighting the need to adopt GMP and HACCP procedures along every step of the milk and cheese production chain in order to reduce the public health risk linked to the consumption of foods contaminated by *B. cereus s.l.*

## 1. Introduction

The *Bacillus cereus* group, also named *B. cereus sensu lato* (*s.l.*), includes numerous closely related species widespread in the environment [1]; among these, *B. cereus* is the most important species from the food safety point of view. As a matter of fact, *B. cereus*, an anaerobic-facultative spore-forming bacteria, is able to survive in the food production environment and subsequently is able to contaminate a wide range of foodstuff: it has been found in milk [2,3,4], dairy products [3,5,6,7], ice cream [8], fresh vegetables [9,10,11], meat [12], spices [13], seafood [14,15], cereal and derivatives [16], and rice [17]. In addition, its capability to form biofilm and viscous heat-resistant spores makes this organism very difficult to remove from food preparation surfaces and food production environments [18,19].

From 2007 to 2014, 413 foodborne outbreaks were reported to be caused by *B. cereus* (*s.l.*), accounting for 6657 cases and 352 hospitalizations [20]. According to the EFSA/ECDC during 2018, 2019, 2020 and 2021, 1539, 1636, 835, and 679 cases, 98, 155, 71, and 87 outbreaks, 111, 44, 10, and 9 hospitalisations and one, seven, one, and one deaths were reported, respectively [2,21,22,23,24].

*B. cereus* can cause two main types of human foodborne disease: a diarrheal and an emetic illness [25,26]. Furthermore, it has been reported that *B. cereus* causes serious extra-intestinal infections, such as severe eye infections, osteomyelitis, hepatitis and inflammatory responses [27,28] and even death [29,30]. The pathogenicity of *B. cereus* is strictly associated with the production of different exotoxins, including pore forming non-hemolytic enterotoxin (NHE), hemolysin BL (HBL), cytotoxin K (*CytK*), enterotoxins FM (*entFM*) and S (*entS*) [31,32] and the emetic toxin cereulide, which is synthesized by non-ribosomal peptide synthetases encoded by the *ces* gene cluster [33]. Unlike the enterotoxins that are synthetized in the intestine after the ingestion of the organism [34,35], the cereulide is a thermo-resistant, proteolysis-resistant, and acid-stable dodecadepsipeptide [19,36] and it is synthesized in contaminated foods.

During the period between 2015 and 2018, in Italy, about 1,200,000 t of milk and about 130,000 t of cheeses were produced each year [37]; despite this large production and the popularity of a lot of traditional well-known cheeses, the foodborne risk associated with *B. cereus* has not been thoroughly evaluated. The aims of this study were: (i) to assess the prevalence of *B. cereus* in milk and dairy products from the Apulia region in southern Italy; (ii) to investigate the genetic diversity of the isolates; and (iii) to assess the potential pathogenicity of the isolates.

## 2. Materials and Methods

### 2.1. Sample Collection

During the period between 2016 and 2017 a total of 550 samples of milk and cheese products in Apulia region (southern Italy) were sampled at retail level, comprising 390 samples of hard cheese (*Caciocavallo*), 128 samples of soft cheeses (83 samples of *mozzarella* and 45 samples of *ricotta* cheese), 27 samples of raw milk and five samples of pasteurized milk. All samples belonged to different manufacturing lots.

### 2.2. Detection and Isolation of B. cereus (s.l.)

The detection of *B. cereus s.l.* was performed by the protocol reported in the European standard ISO 21871:2006 [38]. The qualitative detection was performed using the Mannitol Egg Yolk Polymyxin Agar (MYP) (Biolife Italiana srl—Milan, Italy) as solid mediumincubated at 30 °C for 18–24 h. Suspect colonies were confirmed by the haemolysis test.

### 2.3. Multi-Locus Sequence Typing (MLST)

One isolate per positive sample was characterized by MLST. Genomic DNA was extracted from the *B. cereus* isolates by the DNeasy Blood and Tissue Kit (Qiagen, Hilden, Germany), according to manufacturer’s protocol. Seven housekeeping genes were amplified with different primers and conditions, according to the MLST protocol for *B. cereus* in the PubMLST database [39]. The PCR products were purified using ExoSAP-IT according to the supplier’s recommendations (GE Healthcare, Chicago, IL, USA). Sequence reactions were carried out using BigDye 3.1 Ready reaction mix (Life Technologies, Thermo Fischer Scientific Carlsbad, CA, USA) according to the manufacturer’s instructions. The sequenced products were separated with a 3130 Genetic Analyzer (Life Technologies, Thermo Fischer Scientific Carlsbad, CA, USA). Sequences were imported and assembled with Bionumerics 7.6 software (Applied Maths, Saint Marteen-Latem, Belgium). Any new alleles and STs were assigned by submitting the DNA sequences to the *B. cereus* MLST database].

### 2.4. Detection of Enterotoxin and Emetic Genes

PCR amplification was performed to detect ten genes (*hblA*, *hblC*, *hblD*, *nheA*, *nheB*, *nheC*, *cytK-2*, *entFM*, *entS* and *ces*) with a 25 µL reaction mixture consisting of ~50 ng of genomic DNA, 12.5 µL PCR Buffer, 2 U of enzyme Taq Polymerase, 0.2 mM dNTPs mix and 1 µM each primer. All the primers used in this study are listed in Table 1. The PCR amplifications were performed as previously described [32,40,41,42,43,44]. The data were combined in Microsoft Excel and uploaded to BioNumerics version 7.6 (Applied Maths, Saint Marteen-Latem, Belgium).

### 2.5. Statistical Analysis

STs versus toxin gene profile were compared by Spearman correlation (GraphPad Prism 8.1.1).

## 3. Results

### 3.1. Detection Rate of B. cereus s.l.

*B. cereus s.l*. was detected in 139 of 550 (25.3%) of the analysed samples (Table 2): out of 139 positive samples, three (2.1%) were in raw milk, one (0.7%) was in pasteurized milk and 135 (97.1%) were in cheeses. Among cheeses, *B. cereus s.l.* was detected in: 83 (59.7%) samples of *Cacio cavallo*, 36 (25.9%) samples of *mozzarella* cheese and 16 (11.5%) samples of *ricotta* cheese. The highest contamination rate was detected in *mozzarella* cheese (36 positive samples/83 analysed samples; 43.3%), the lowest in raw milk samples (three positive samples/27 analysed samples; 1.1%). A contamination rate of 21.2% (83/390), 35.5% (16/45) and 20% (1/5) were detected in *Caciocavallo*, *ricotta* cheese and pasteurized milk, respectively.

### 3.2. Strains Characterization by MLST

The results of the isolates characterization are summarized in Table 2 and Figure 1. A total of 139 isolates were characterized by the MLST and 73 STs were assigned; 12 isolates were assigned to new STs. Twenty-two isolates were grouped into nine clonal complexes (CCs). The dominant STs among all positive samples were ST 1986 and ST 34, instead, ST34, ST12 and ST26 were the dominant STs in *Caciocavallo*, *ricotta* cheese and *mozzarella*, respectively. The new detected STs were: ST2491, ST2667, ST2660, ST2661, ST2683, ST2664, ST2669, ST2666, ST2671, ST2675, ST2670 and ST2682.

### 3.3. Distribution of Enterotoxin Genes

The distribution of virulence genes is reported in Table 2 and Figure 2. According to the pathogenic characteristics of *B. cereus s.l*. the virulence genes were divided into two categories, namely enterotoxin coding genes (*hblACD*, *nheABC*, *cytK*, *entFM*, *entS*) and cereulide synthetase gene (*ces*). Among the enterotoxin genes, the genes of the *nhe*ABC cluster were detected in 90% of the isolates and in most of them was recovered the entire *nhe*ABC gene cluster; only three isolates were lacking the entire *nheABC* gene cluster. The gene cluster *hblCDA* was found in 61% of isolates, with a frequency not much different among the three genes detected: 68% of isolates harboured the *hblA* gene, 64% of isolates harboured the *hblC* gene and 75% of isolates harboured the *hblD* gene. The *entFM* and *entS* genes were detected in 94% of isolates; the *cytK* gene was identified in 44%. The *ces* gene was detected in 8% of the total isolates. Based on the distribution of both, enterotoxin and emetic genes, we divided our isolates in thirty-eight (1-38) different genetic profiles (Table 2).

### 3.4. Statistical Analysis

When we compared the STs versus toxin gene profile by Spearman correlation (GraphPad Prism 8.1.1), no statistical significance was shown (*r* = −0.01462; *p* = 0.87144).

## 4. Discussion

*B. cereus* is a ubiquitous spore-forming bacterium widespread in the environment and frequently detected in a wide range of foodstuffs, including milk and cheeses, where it can act as foodborne pathogen. Because of the ubiquitous presence of *B. cereus* spores, dairy products can become contaminated during milking, processing or storage stages [45,46]. As a matter of fact, during 2007–2014, dairy products were responsible for 11 outbreaks due to *B. cereus* registered in EU [20]. The aim of our survey was to investigate the presence of *B. cereus s.l.*, the genetic profile and the distribution of genetic virulence markers in the isolates obtained from milk (raw and pasteurized) and traditional cheeses from the Apulia region, southern Italy. In our study we found that 25.3% of all samples were contaminated by *B. cereus s.l.*; to the best of our knowledge, in Italy, data on the prevalence and the genetic characterization of *B. cereus* in milk and dairy products are scarce, thus it is not possible to make any comparison with previous reports. Other countries reported similar results that we found here: in China [2,47], Abidjan [48] and Brazil [49], *B. cereus* was detected in 27% of the milk and dairy products analysed. Recently, Adame-Gomez and colleagues reported a frequency of 29.5% of *B. cereus* contamination in artisanal cheese made with raw milk product in Mexico [50], whereas, in Ghana [3], Slovakia [51], India [4] and Northeast China [52], *B. cereus* was reported with a contamination level that ranged from 14 to 55%. Notably, among the food samples considered in this survey, we identified *B. cereus s.l.* in pasteurized milk; considering the ability of *B. cereus* to form spores that are able to survive heat treatment commonly applied in milk pasteurization [53,54], this result is not surprising. Although it is not possible to ascertain if the contamination occurs before, during the pasteurization or the filling process, spore germination and vegetative cells replication is a remote possibility, given that pasteurized milk must be stored under refrigeration and also considering its short shelf-life. In addition, psychrotrophic strains of *B. cereus* are known, but these isolates are generally recognized as not able to produce cereulide, thus the presence of this contamination should not pose a risk in refrigerated foods [55].

We analysed 390 samples of *Caciocavallo*, and 21.2% of these were contaminated by *B. cereus s.l.*. *Caciocavallo* is a very appreciated traditional curd-stretching cheese which has a large distribution in southern Italy, where it is produced from raw or pasteurized bovine milk; the season can last from two to six or more months [56].

In our survey, *mozzarella* and *ricotta* (soft cheeses) resulted contaminated in 25.8% and 11.5%, respectively, among all samples considered. These soft cheeses showed the highest rate of contamination compared other samples, with a contamination rate of 43.3% for *mozzarella* and 35.5% for *ricotta*. *Mozzarella* is the most famous Italian curd-stretching cheese, produced from raw or pasteurized cow or water buffalo milk [56]; in fact, some *mozzarella* cheeses have been awarded the protected designation of origin (POD). *Ricotta* is a much-appreciated food which is largely employed for direct consumption or as an ingredient for bakery and pastries products. The production process of this unripened acid-heat coagulated dairy originating from the whey obtained from draining cheese curds from curdling of other cheeses [57], includes a step in which it is applied a temperature that is able to reduce the pathogenic vegetative flora [58]. However, spore-forming bacteria could survive at these steps and, if other favourable conditions occur, they could germinate and multiply causing an active infection or an intoxication. In addition, it is not possible to establish whether the contamination occurs in the *pre-* or *post-*process steps, considering the heath resistance of *B. cereus* spores. Our results showed for the first time the high prevalence of *B. cereus s.l.* in milk and dairy products from Italy; it is known that dairy products, that are subjected to a mild heat treatment and are characterized by an extended shelf life under refrigeration, could be at risk because the competitive vegetative microflora is inactivated either by heat treatment or by reduced water activity, but enables the survival of the spores of *B. cereus* [53]. The high prevalence identified in our study supports the potential risk due to the consumption of *B. cereus*-contaminated dairy products; despite the existence of *B. cereus* psychrotolerant isolates capable to produce emetic toxin at low temperature [59], the main management option for controlling the growth of the organism in foodstuff remains the chill chain (storage temperature between <4 °C and 7 °C) that could be maintained during the entire production until the consumers receive them [49,54].

In our study, we did not assess the quantification of the samples contamination; on the other hand, it is known that it is very difficult to estimate the number of Colony Forming Unit (CFU) of *B. cereus* per g of food that could represent a risk for consumers: in literature cases in which 10^3^ CFU/g were responsible of diseases are reported [20], although a concentration of *B. cereus* above 10^5^ CFU/g is considered hazardous [60,61,62]. In the EU food law, *B. cereus* is considered as Process Hygiene Criterion for dried infant formulae and dried dietary foods for medical purposes intended for infants below six months of age; in case of exceeding the established limits (between 50 and 500 CFU/g), the food operator must improve his hygiene production procedures and prevent recontamination. In addition, they must carry out a selection of the raw material which has been used [63].

Based on the MLST profile, our isolates showed a high genetic diversity. Indeed, 12 of 73 MLST profiles (16.4% of the studied strains) were detected for the first time in this study; whereas the remaining profiles were already present in the MLST *B. cereus* database and, among these, twenty-three STs were previously identified in milk and milk products [64,65,66]. The other STs were previously identified in other foods or in the environment [65,67,68,69,70]. *B. cereus* isolates related with emetic food poisoning, were generally associated with ST26 [33,42,43,71,72]; in our study, among the four isolates belonging to ST26, three harboured *ces* gene, supporting the hypothesis that this genotype frequently harbour the potential emetic strains; however, these strains did not carry the *hbl* cluster gene, as was already reported [33]. Interestingly, we identified nine additional STs (ST4, ST33, ST142, ST1936, ST2031, ST2045, ST2062, ST2667 and ST2666) that may become important to define the potential emetic strains for the presence of *ces* gene. Moreover, we noticed that all the isolates belonging to the CC18 and CC205 harboured the same virulence profile (profile 2). Furthermore, we identified an isolate that carried the *ces* gene and among the enterotoxin genes we detected only the *entS* gene. Based on genetic profile, we supposed that the infection due to the two isolates belonging to profile 1 (ST4; ST33) could be associated to a worst-case scenario [73,74], since they harboured all the genes investigated. The genetic profiles 2 and 5 represented the largest group, which included a total of 39 and 27 isolates, respectively. The isolates that belonged to genetic profile 2, including 19 STs, harboured all the enterotoxigenic genes, suggesting their enterotoxigenic potential; whereas the isolates that belonged to genetic profile 5, that included 21 STs, harboured eight of the nine enterotoxin genes, since they did not carry the *cytK* gene. This finding confirms that the health risk for consumers is strongly strain dependent, owing to the various pathogenic potential [20].

*B. cereus* diarrhoeal disease is due to the synthesis of different enterotoxins in the small intestine [29]; according to EFSA, about all strains of *B. cereus* carry the non-haemolytic enterotoxin complex genes (*nhe*), whereas about 30–70% of the isolates harboured the *hbl* and *cytK-2* genes [20]. In our study, nine enterotoxigenic genes were detected and the proportion of *hblACD*, *nheABC* and *cytK* genes in *B. cereus s.l.* isolates were found to be, 61%, 90% and 44%, respectively. These frequencies were similar to previous studies [2,75,76,77], suggesting that the diarrheagenic isolates have a high distribution among foods. Moreover, we detected two other enterotoxin genes (*entFM* and *entS*), which contribute to the severity of diarrheal illness [78]. Differently, the emesis is due to the heat-stable dodecadepsipeptide cereulide, that, similarly to other reports [3,79] we observed in 8% of the isolates. A number of studies failed to detect *ces* genes from food isolates [77,80], probably because the emetic isolates of *B. cereus* belong to a specific lineage that has a different geographical circulation [81]; however, it is difficult to give a conclusive opinion about the potential pathogenicity of the isolates because it is known that a large variability in synthesis of these enterotoxins exist among the *B. cereus* isolates, as well as it is still unknown why isolates that show the same genetic profile show differences in the amount of toxin production [19,47]. The strain’s growth and the cereulide production are due to the composition of the food and the environment in different way; Ellouze recently demonstrated that in some food matrices (i.e., cereal-based foods) the growth rate of *B. cereus* is smaller than in other foods (i.e., milk-based products) but the cereulide production is faster than in foods where the growth is faster [82].

## 5. Conclusions

We believe that the potential hazard associated with contaminated foods by *B. cereus s.l.* should not be ignored. To ensure the protection of the public health, it is therefore necessary to develop procedures to prevent contamination of raw ingredients, to implement a plan based on hazard analysis and critical control point procedures during the manufacturing process of dairy products and strictly to respect the chill chain during their commercial and home storage. 

The lack of information about the acidity, salinity, water activity, moisture and the cultures used for the production of the samples we analysed, factors that affect the growth of *B. cereus*, represent a weakness in our survey; further studies aimed at correlating these critical factors with the growth and toxin production by *B. cereus* in such foods are needed [44].

## Figures and Tables

**Figure 1 foods-12-01548-f001:**
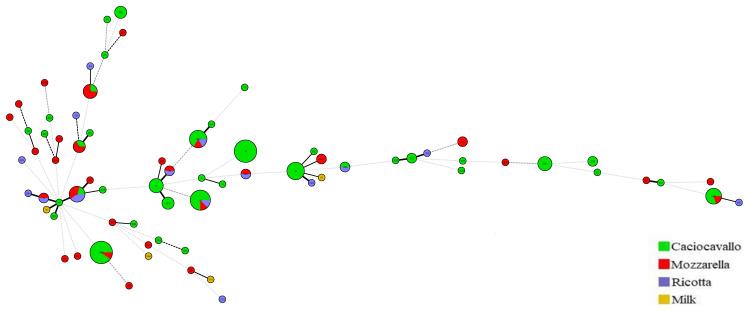
Minimum spanning tree analysis, using BioNumerics version 7.1, of the 139 *B. cereus* isolates based on allelic profiles of seven housekeeping genes. Each circle corresponds to a sequence type (ST) and the size of the circle is related to the number of isolates found with that profile. Each colour inside the circles represents the source origin of the strains (green, *Caciocavallo*; red, *Mozzarella*; indigo, *Ricotta*; yellow, milk).

**Figure 2 foods-12-01548-f002:**
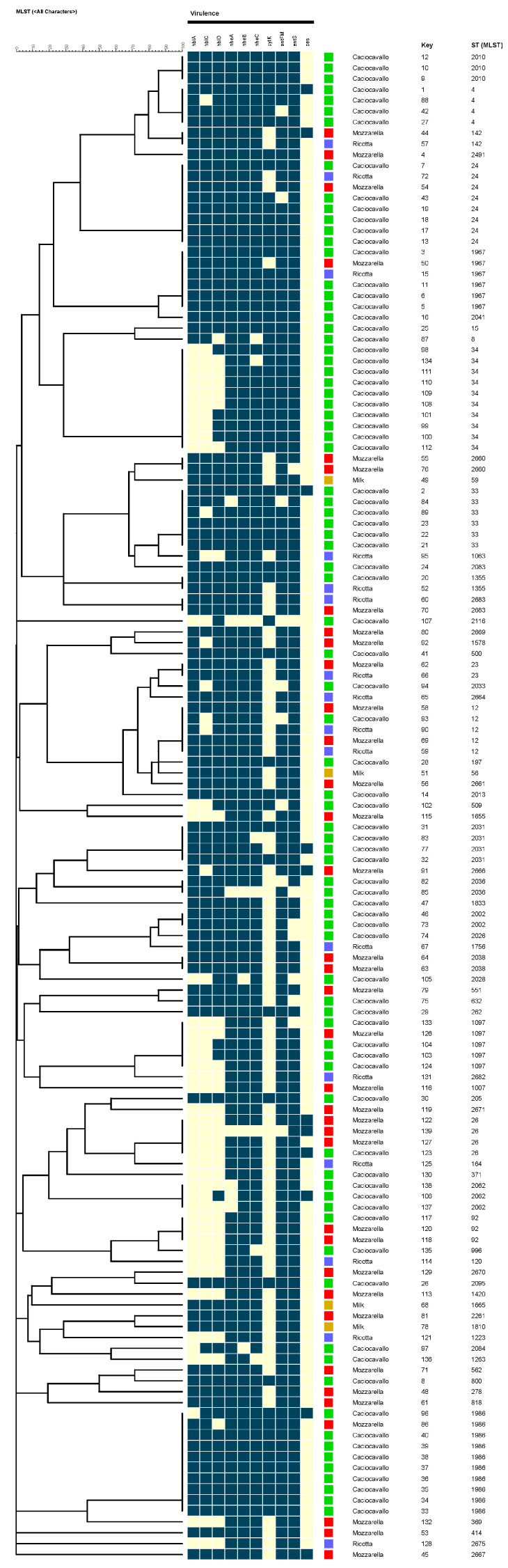
Bionumerics phylogenetic tree built on 139 *B. cereus* isolates. The tree is based on 73 ST profiles. The presence of virulence factors are marked in blue colour. The source and STs are listed next the tree.

**Table 1 foods-12-01548-t001:** PCR primers targeting the virulence genes investigated in this study.

Target	Gene	Primer Sequence (5′-3′)	Reference
*hblA*	hblA_F	GTGCAGATGTTGATGCCGAT	[40]
	HBLA_R	ATGCCACTGCGTGGACATAT	
*hblC*	L2A_F	AATGGTCATCGGAACTCTAT	[40]
	L2B_R	CTCGCTGTTCTGCTGTTAAT	
*hblD*	L1A_F	AATCAAGAGCTGTCACGAAT	[40]
	L1B_R	CACCAATTGACCATGCTAAT	
*nheA*	nheA_F	TACGCTAAGGAGGGGCA	[40]
	nheA_R	GTTTTTATTGCTTCATCGGCT	
*nheB*	nheB_F	CTATCAGCACTTATGGCAG	[32]
	nheB_R	ACTCCTAGCGGTGTTCC	
*nheC*	nheC_F	CGGTAGTGATTGCTGGG	[32]
	nheC_R	CAGCATTCGTACTTGCCAA	
*cytK*	cytK_F	ACAGATATCGGKCAAAATGC	[26]
	cytK_R	TCCAACCCAGTTWSCAGTTC	
*entFM*	entFM_F	ATGAAAAAAGTAATTTGCAGG	[41]
	entFM_R	TTAGTATGCTTTTGTGTAACC	
*entS*	entS_F	GGTTTAGCAGCAGCTTCTGTAGCTGGCG	[41]
	entFM_R	CTTGTCCAACTACTTGTAGCACTTGGCC	
*ces*	ces_F	GGTGACACATTATCATATAAGGTG	[42]
	ces_R	GTAAGCGAACCTGTCTGTAACAACA	

**Table 2 foods-12-01548-t002:** Characterization and virulence genetic profile of the *B. cereus s.l.* isolated from cheeses and milk. The STs identified for the first time were indicated in bold.

Source	ST	CC	Genes Encoding for Enterotoxins									Cereulide Encoding Gene	Genetic Profile
			*hblA*	*hblC*	*hblD*	*nheA*	*nheB*	*nheC*	*cytK*	*entFM*	*entS*	*ces*	
Caciocavallo	4	ST-142 complex	+	+	+	+	+	+	+	+	+	+	1
Caciocavallo	33		+	+	+	+	+	+	+	+	+	+	1
Caciocavallo	1967		+	+	+	+	+	+	+	+	+	-	2
Mozzarella	**2491**		+	+	+	+	+	+	+	+	+	-	2
Caciocavallo	1967		+	+	+	+	+	+	+	+	+	-	2
Caciocavallo	1967		+	+	+	+	+	+	+	+	+	-	2
Caciocavallo	24		+	+	+	+	+	+	+	+	+	-	2
Caciocavallo	800	ST-111 complex	+	+	+	+	+	+	+	+	+	-	2
Caciocavallo	2010	ST-142 complex	+	+	+	+	+	+	+	+	+	-	2
Caciocavallo	2010	ST-142 complex	+	+	+	+	+	+	+	+	+	-	2
Caciocavallo	1967		+	+	+	+	+	+	+	+	+	-	2
Caciocavallo	2010	ST-142 complex	+	+	+	+	+	+	+	+	+	-	2
Caciocavallo	24		+	+	+	+	+	+	+	+	+	-	2
Caciocavallo	2013	ST-23 complex	+	+	+	+	+	+	+	+	+	-	2
Ricotta	1967		+	+	+	+	+	+	+	+	+	-	2
Caciocavallo	2041		+	+	+	+	+	+	+	+	+	-	2
Caciocavallo	24		+	+	+	+	+	+	+	+	+	-	2
Caciocavallo	24		+	+	+	+	+	+	+	+	+	-	2
Caciocavallo	24		+	+	+	+	+	+	+	+	+	-	2
Caciocavallo	1355	ST-18 complex	+	+	+	+	+	+	+	+	+	-	2
Caciocavallo	33		+	+	+	+	+	+	+	+	+	-	2
Caciocavallo	33		+	+	+	+	+	+	+	+	+	-	2
Caciocavallo	33		+	+	+	+	+	+	+	+	+	-	2
Caciocavallo	2083		+	+	+	+	+	+	+	+	+	-	2
Caciocavallo	15	ST-8 complex	+	+	+	+	+	+	+	+	+	-	2
Caciocavallo	2095		+	+	+	+	+	+	+	+	+	-	2
Caciocavallo	4	ST-142 complex	+	+	+	+	+	+	+	+	+	-	2
Caciocavallo	197	ST-23 complex	+	+	+	+	+	+	+	+	+	-	2
Caciocavallo	262		+	+	+	+	+	+	+	+	+	-	2
Caciocavallo	205	ST-205 complex	+	+	+	+	+	+	+	+	+	-	2
Caciocavallo	2031		+	+	+	+	+	+	+	+	+	-	2
Caciocavallo	2031		+	+	+	+	+	+	+	+	+	-	2
Caciocavallo	1986		+	+	+	+	+	+	+	+	+	-	2
Caciocavallo	1986		+	+	+	+	+	+	+	+	+	-	2
Caciocavallo	1986		+	+	+	+	+	+	+	+	+	-	2
Caciocavallo	1986		+	+	+	+	+	+	+	+	+	-	2
Caciocavallo	1986		+	+	+	+	+	+	+	+	+	-	2
Caciocavallo	1986		+	+	+	+	+	+	+	+	+	-	2
Caciocavallo	1986		+	+	+	+	+	+	+	+	+	-	2
Caciocavallo	1986		+	+	+	+	+	+	+	+	+	-	2
Caciocavallo	500		+	+	+	+	+	+	+	+	+	-	2
Caciocavallo	4	ST-142 complex	+	+	+	+	+	+	+	-	+	-	3
Caciocavallo	24		+	+	+	+	+	+	+	-	+	-	3
Mozzarella	142		+	+	+	+	+	+	-	+	+	+	4
Mozzarella	**2667**		+	+	+	+	+	+	-	+	+	+	4
Caciocavallo	2002		+	+	+	+	+	+	-	+	+	-	5
Caciocavallo	1833		+	+	+	+	+	+	-	+	+	-	5
Mozzarella	278		+	+	+	+	+	+	-	+	+	-	5
Milk	59		+	+	+	+	+	+	-	+	+	-	5
Mozzarella	1967		+	+	+	+	+	+	-	+	+	-	5
Milk	56		+	+	+	+	+	+	-	+	+	-	5
Ricotta	1355		+	+	+	+	+	+	-	+	+	-	5
Mozzarella	414		+	+	+	+	+	+	-	+	+	-	5
Mozzarella	24		+	+	+	+	+	+	-	+	+	-	5
Mozzarella	**2660**		+	+	+	+	+	+	-	+	+	-	5
Mozzarella	**2661**		+	+	+	+	+	+	-	+	+	-	5
Ricotta	142		+	+	+	+	+	+	-	+	+	-	5
Mozzarella	12		+	+	+	+	+	+	-	+	+	-	5
Ricotta	12	ST-23 complex	+	+	+	+	+	+	-	+	+	-	5
Ricotta	**2683**		+	+	+	+	+	+	-	+	+	-	5
Mozzarella	818		+	+	+	+	+	+	-	+	+	-	5
Mozzarella	23	ST-23 complex	+	+	+	+	+	+	-	+	+	-	5
Mozzarella	2038		+	+	+	+	+	+	-	+	+	-	5
Mozzarella	2038		+	+	+	+	+	+	-	+	+	-	5
Ricotta	**2664**		+	+	+	+	+	+	-	+	+	-	5
Ricotta	23		+	+	+	+	+	+	-	+	+	-	5
Ricotta	1756		+	+	+	+	+	+	-	+	+	-	5
Milk	1665		+	+	+	+	+	+	-	+	+	-	5
Mozzarella	12		+	+	+	+	+	+	-	+	+	-	5
Mozzarella	**2683**		+	+	+	+	+	+	-	+	+	-	5
Mozzarella	562	ST-111 complex	+	+	+	+	+	+	-	+	+	-	5
Ricotta	24		+	+	+	+	+	+	-	+	+	-	5
Caciocavallo	2002		+	+	+	+	+	+	-	+	-	-	6
Caciocavallo	2026		+	+	+	+	+	+	-	+	-	-	6
Caciocavallo	632		+	+	+	+	+	+	-	+	-	-	6
Mozzarella	**2660**		+	+	+	+	+	+	-	+	-	-	6
Caciocavallo	2031		+	+	+	+	+	+	-	+	+	+	7
Milk	1810		+	+	+	+	+	+	-	+	+	-	8
Mozzarella	551		+	+	+	+	+	+	-	+	+	-	8
Mozzarella	**2669**		+	+	+	+	+	+	-	+	+	-	8
Mozzarella	2261		+	+	+	+	+	+	-	+	+	-	8
Caciocavallo	2036		+	+	+	+	+	+	-	-	+	-	9
Caciocavallo	2031		+	+	+	+	+	-	-	+	+	-	10
Caciocavallo	33		+	+	+	-	+	+	+	-	+	-	11
Caciocavallo	2036		+	+	+	-	-	-	-	+	-	-	12
Mozzarella	1986		+	+	-	+	+	+	+	+	+	-	13
Caciocavallo	8	ST-8 complex	+	+	-	+	+	-	+	+	+	-	14
Caciocavallo	4	ST-142 complex	+	-	+	+	+	+	+	+	+	-	15
Caciocavallo	33		+	-	+	+	+	+	+	+	+	-	15
Ricotta	12		+	-	+	+	+	+	-	+	+	-	16
Mozzarella	**2666**		+	-	+	+	+	+	-	+	+	+	17
Mozzarella	1578	ST-97 complex	+	-	+	+	+	+	-	+	+	-	18
Caciocavallo	12	ST-23 complex	+	-	+	+	+	+	-	-	+	-	19
Caciocavallo	2033	ST-23 complex	+	-	+	+	+	+	-	-	+	-	19
Ricotta	1063		+	-	-	+	+	+	-	+	+	-	20
Caciocavallo	1986		-	+	+	+	+	+	+	+	+	+	21
Caciocavallo	2084	ST-365 complex	-	+	+	+	-	+	-	+	+	-	22
Caciocavallo	34		-	-	+	+	+	+	+	+	+	-	23
Caciocavallo	34		-	-	+	+	+	+	+	+	+	-	23
Caciocavallo	34		-	-	+	+	+	+	+	+	+	-	23
Caciocavallo	34		-	-	+	+	+	+	+	+	+	-	23
Caciocavallo	509		-	-	+	+	+	+	+	-	+	-	24
Caciocavallo	1097		-	-	+	+	+	+	-	+	+	-	25
Caciocavallo	1097		-	-	+	+	+	+	-	+	+	-	25
Caciocavallo	2028		-	-	+	+	-	+	-	+	-	-	26
Caciocavallo	2062		-	-	+	-	+	+	-	+	+	+	27
Caciocavallo	2116		-	-	+	-	-	-	+	-	-	-	28
Caciocavallo	34		-	-	-	+	+	+	+	+	+	-	29
Caciocavallo	34		-	-	-	+	+	+	+	+	+	-	29
Caciocavallo	34		-	-	-	+	+	+	+	+	+	-	29
Caciocavallo	34		-	-	-	+	+	+	+	+	+	-	29
Caciocavallo	34		-	-	-	+	+	+	+	+	+	-	29
Mozzarella	1420		-	-	-	+	+	+	-	+	+	-	30
Ricotta	120		-	-	-	+	+	+	-	+	+	-	30
Mozzarella	1655		-	-	-	+	+	+	-	+	+	-	30
Mozzarella	1007		-	-	-	+	+	+	-	+	+	-	30
Caciocavallo	92		-	-	-	+	+	+	-	+	+	-	30
Mozzarella	92		-	-	-	+	+	+	-	+	+	-	30
Mozzarella	**2671**		-	-	-	+	+	+	-	+	+	-	30
Mozzarella	92		-	-	-	+	+	+	-	+	+	-	30
Ricotta	1223		-	-	-	+	+	+	-	+	+	-	30
Mozzarella	26		-	-	-	+	+	+	-	+	+	+	31
Caciocavallo	26		-	-	-	+	+	+	-	+	+	+	31
Caciocavallo	1097		-	-	-	+	+	+	-	+	+	-	32
Ricotta	164		-	-	-	+	+	+	-	+	+	-	32
Mozzarella	1097		-	-	-	+	+	+	-	+	+	-	32
Mozzarella	26		-	-	-	+	+	+	-	+	+	-	32
Ricotta	**2675**		-	-	-	+	+	+	-	+	+	-	32
Mozzarella	**2670**		-	-	-	+	+	+	-	+	+	-	32
Caciocavallo	371		-	-	-	+	+	+	-	+	+	-	32
Ricotta	**2682**		-	-	-	+	+	+	-	+	+	-	32
Mozzarella	369		-	-	-	+	+	+	-	+	+	-	33
Caciocavallo	1097		-	-	-	+	+	+	-	+	-	-	34
Caciocavallo	34		-	-	-	+	+	-	+	+	+	-	35
Caciocavallo	996		-	-	-	+	+	-	-	+	+	-	36
Caciocavallo	1263	ST-365 complex	-	-	-	+	+	-	-	+	+	-	36
Caciocavallo	2062		-	-	-	-	+	+	-	+	+	-	37
Caciocavallo	2062		-	-	-	-	+	+	-	+	+	-	37
Mozzarella	26		-	-	-	-	-	-	-	-	+	+	38

## Data Availability

Data is contained within the article.

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
