# Peer review of "High Genetic Diversity and Virulence Potential in Bacillus cereus sensu lato Isolated from Milk and Cheeses in Apulia Region, Southern Italy"

_foods, 2023, doi:10.3390/foods12071548_

Round 1

Reviewer 1 Report

This is an interesting and meaningful survey-type study. The safety/toxicity information is important. Although the samples are collected within one country (Italy), the results probably have a wide relevance. 

A major question is about the data analysis and also the apparent lack of important sample quality parameters that have a significant impact on microbial/pathogen growth. For example, the acidity, salinity, mositure content, water activity, and the cultures used in the production of different chesses and milk, i.e., 390 hard cheese (Caciocavallo), 128 soft cheeses (83 mozzarella and 45 ricotta cheese), 27 raw milk and pasteurized milk) are inherent quality/safety factors. These values should be tested and included in the statistical model to generalize the susceptibility of milk products to B. cereus growth and potential toxins formation. 

Author Response

Dear Reviewer,

The authors are grateful to you for your comments and suggestions.

English language and style:

  • The text has been revised and improved by a native English speaker, to make improvements to it.
  • The Introduction section has been improved and new references have been included.
  • The Method section has been improved with some specifications on the Detection and isolation of the studied strains.
  • The Results section has been improved and made clearer through some changes in Figure 1.
  • In order to improve clarity, the Conclusions section has been reported as a separate section.

As far as regards the comments of the reviewer, the authors agree with the importance to known the physical-chemical and microbial characteristics of the samples in order to make a forecast regarding the growth of B. cereus and the potential toxin production. However, these aspects were not included in the aims of the work, which represents a first step of a comprehensive study on B. cereus. In this part of the study, the authors had investigated the prevalence of B. cereus in milk and in well-known and consumer dairy products and the bio-molecular characteristics of the isolates. Further investigations aimed to correlate the quality parameters of dairy products with the growth of B. cereus and the toxin production, are warranted.

Reviewer 2 Report

Dear authors,

the manuscript presents an interesting study.

Some minor aspects need to be adressed:

Figure 1 and 2 needs quality improvements.

3.4. Statistical analysis - there is no further discussions and presentation e.g. table or figure to show the results.

Line 186 - mentioned southern Italy, therefore please correct as well in line 57, as it is specifically a region of Italy.As well i would suggest same modification in the title, and please mention about region in the abstract.

Conclusions section is completly missing. Lines 287-293 shall be included in the separate conclusions section.

Author Response

Dear Reviewer,

Thanks a lot for your comments and suggestions. The authors greatly appreciated it.

1) The text has been revised and improved by a native English speaker, to make improvements to it.

2) The figure has been revised to improve its  quality, as suggested.

3) The results of the statistical analysis have not been reported in a table form because they have shown no statistical significance.

4) According to the reviewer suggestion, the authors have corrected the Title, the Abstract, and the text, also adding and specifying the Region in which the survey has been conducted.

5) The Conclusion has been reported as a separate section, as suggested by the reviewer

Round 2

Reviewer 1 Report

The authors have made significant modifications of the text. However, their response to the question of samples' acidity, salinity, moisture content, water activity, and type of cultures used in the production is not satisfactory (elusive). Essentially authors did not answer the question. If authors could not provide such critical factors, which are known to affect the growth of B. cereus, due to an oversight, they should at least acknowledge this weakness of their experimental design. An appropriate place to indicate such factors, which should have been considered but were not included in the present survey, is the Conclusion section. Please state that these factors should be included in follow up or future studies.  
